# Characterization of a Translucent Material Produced from *Paulownia tomentosa* Using Peracetic Acid Delignification and Resin Infiltration

**DOI:** 10.3390/polym14204380

**Published:** 2022-10-17

**Authors:** Kyoung-Chan Park, Byeongho Kim, Hanna Park, Yesun Kim, Se-Yeong Park

**Affiliations:** Department of Forest Biomaterials Engineering, Kangwon National University, Chuncheon 24341, Korea

**Keywords:** translucent wood, *Paulownia tomentosa*, peracetic acid delignification, transmittance and haze, bending strength

## Abstract

*Paulownia tomentosa*, a tree species that allows for efficient production of translucent wood, was selected as an experimental wood species in this study, and a two-step process of delignification and polymer impregnation was performed. For delignification, 2–4 mm thick specimens were immersed in peracetic acid for 8 h. The delignified-wood specimens were impregnated using epoxy, a commercial transparent polymer. To identify the characteristics of the resulting translucent wood, the transmittance and haze of each type of wood section (cross- and tangential) were measured, while bending strength was measured using a universal testing machine. The translucent wood varied in properties according to the wood section, and the total transmittance and haze were 88.0% and 78.5% for the tangential section and 91.3% and 96.2% for the cross-section, respectively. For the bending strength, untreated wood showed values of approximately 4613.5 MPa modulus of elasticity (MOE), while the epoxy impregnation to improve the strength of the wood had increased the MOE up to approximately 6089.9 MPa, respectively. A comparative analysis was performed in this study with respect to the substitution of balsa, which is used widely in the production of translucent wood. The results are anticipated to serve as baseline data for the functionalization of translucent wood.

## 1. Introduction

Wood is a renewable material that is most abundantly found on Earth. It is a natural polymer with a complex chemical structure of cellulose fibers surrounded by hemicellulose and lignin [1,2]. A method to isolate cellulose and lignin from wood was first proposed in 1838 [3]. Delignification is a process to remove lignin and a portion of hemicellulose from wood to modify the chemical composition and reduce the mechanical strength. Hence, delignified wood that has lost its original form of porous and hierarchical structure is pulverized to wood powder or chips for use in pulp manufacture and biorefinery. However, delignification performed at 80–120 °C and atmospheric pressure does not significantly affect the wood structure such that the original 3D structure is retained [4]. It also becomes easier to induce hydroxyl groups in cellulose to confer functionality [5].

Various studies have been conducted regarding the use of wood and wood components, and the study on the use of delignified wood was initiated in 1992. Fink first proposed a method to create translucent wood through the delignification of wood to produce a porous material with the cell wall structure preserved, followed by transparent polymer impregnation [6]. The method to produce translucent wood was further specified by the Kungliga Tekniska högskolan (KTH) in Sweden in 2016. Li et al. (2016) discussed in detail the preparation of translucent wood and the results for its optical and mechanical performance, thus providing the basis for numerous subsequent studies [7]. Among the various ongoing studies based on Li et al. (2016) are the one on the window for which the transmittance and haze can be controlled through polymer impregnation and lamination with a color-changing property using electricity and the one on the translucent wood modified to display the tree ring [8,9]. In addition to the production of translucent wood, the use of delignification to improve the insulation performance of wood for application as an excellent insulator has been reported [10,11]. Delignified wood exhibits a high potential for application in the development of advanced composite materials with various other functional materials, which will continue to be pursued in further studies.

Among the wood species to produce delignified wood, *Ochroma pyramidale* (balsa tree) has been considered a representative wood. It is well-known as a large and fast-growing tree native to the Americas and with low specific gravity in the range of 0.1–0.2; thus, it has advantages when proceeding with the functionalization and resin impregnation [7,12].

Meanwhile, the Intergovernmental Panel on Climate Change’s (IPCC) guideline acknowledges the storage effect of delayed carbon emission via the consumption of harvested wood products within respective countries [13]. For making wood-based functional materials, finding suitable wood species as alternatives to the balsa tree in domestic settings is needed.

*Paulownia tomentosa*, a tree species widely distributed in South Korea as well as in other countries in Asia, was selected as the experimental wood species as an alternative to *O. pyramidale*. The specific gravity of *P. tomentosa* is the lowest in Asia, with values in the range of 0.2–0.3, while it is a fast-growing tree species with high porosity. In this study, delignification was applied to *P. tomentosa*, and the potential of translucent wood was evaluated by impregnating the pores with increased size accordingly with transparent epoxy. The ultimate goal is to create a new material that can replace glass by synthesizing a biodegradable transparent polymer with wood.

## 2. Materials and Methods

### 2.1. Materials

In this study, *Paulownia tomentosa*, which grows in the academic forest of Kangwon National University in Hongcheon-gun, Gangwon-do, was used for experimental wood specimens. To compare the optical properties according to the wood section, the samples were prepared by dividing into cross- and tangential sections. The cross-section specimen was cut to a size of 2 (L) × 30 (W) × 40 (T) mm^3^, and the tangential-section specimen was cut to a size of 30 (L) × 2 (W) × 40 (T) mm^3^.

### 2.2. Peracetic Acid Delignification

#### 2.2.1. Peracetic Acid Preparation

Acetic acid (99%, Daejung, Korea) and hydrogen peroxide (HP, 30%, Daejung, Korea) were mixed in a 1:1.5 (*v*/*v*) ratio to make the peracetic acid (PAA) as a delignification agent. After mixing, the solution was sufficiently mixed through stirring for 1 h. The concentrations of PAA and HP were measured according to the titration method [14]. After storage at room temperature for 14 days, the concentrations were of 6.84% PAA and 10.88% HP, respectively.

#### 2.2.2. Delignification

For delignification, three *P. tomentosa* specimens (total of about 4.5 g) and 250 mL of peracetic acid were placed in a 1 L beaker and then delignified with peracetic acid for 8 h. After washing off the decomposed residual lignin using distilled water, ethanol was treated from a low concentration to a high concentration to proceed with dehydration. Finally, the specimens were stored in acetone for acetylation.

### 2.3. Transparent Polymer Impregnation

For polymer infiltration, crystal resin (Crysin 2.5, Marvel Epoxy, Korea) was purchased as an epoxy polymer. A resin was prepared by mixing the main agent and the curing agent in a ratio of 4:1 (*w*/*w*). Delignified-wood specimens (total of about 20 g) were administered to 400 g of the prepared epoxy, and impregnation was performed using a vacuum desiccator (VDP-30UG, Lab companion, Korea). The vacuum desiccator’s pressure was maintained at 0.1 MPa, and the pressure was repeatedly reduced three times. The impregnated wood was dried at 25 °C for 24 h in a place not exposed to direct sunlight to produce translucent wood.

### 2.4. ATR FT-IR Spectroscopy Analysis

Fourier-transform infrared spectroscopy (FT-IR, Nicolet Summit, Thermo Fisher Scientific, Waltham, MA, USA) was used to observe the functional group changes of wood specimens according to chemical treatment. The spectra were recorded in the wavenumber range from 4000–800 cm^−1^, and a resolution of 4 cm^−1^ per sample with 32 scans were used.

### 2.5. Lignin and Sugar Content

The content of insoluble lignin (klason lignin) was measured according to TAPPI T222 om-02. The content of acid-soluble lignin was analyzed by UV–Visible spectrometer (UV-VIS, Optizen pop s, KLab, Daejeon, Korea) of the filtrate according to TAPPI UM 205.

Sugar content analysis was performed using a bio-liquid chromatograph (ICS-5000, Thermo Dionex, Palo Alto, CA, USA). The column was CarboPac PA-1 (250 × 4 mm, Dionex, Palo Alto, CA, USA), and the detector was a pulsed amperometry detector (HP 1100, Hewlett Packard, USA). The assay was performed with potassium hydroxide (1–35 min: 2 mM; 35–36 min: 2 → 100 mM; 36–56 min: 100 mM; 56–57 min: 100 → 2 mM; 57–63 min: 2 mM) at 1 mL/min, and the injection amount was 10 μL. For standard substances, a calibration curve was prepared using glucose, xylose, arabinose, galactose, and rhamnose.

### 2.6. Observation of Cell Wall Structure

SEM (CX-200TM, COXEM, Daejeon, Korea) analysis was performed to compare the cell wall structures of untreated, peracetic acid-treated, and epoxy-impregnated wood. The observation sample was prepared in a size of 5 × 5 × 5 mm^3^. Afterwards, the sample was coated with about 10 nm of platinum using an ion sputtering coater (Leica, EM, ACE600) and then photographed with an acceleration voltage of 20 kV.

### 2.7. Transmittance Measurement

Transmittance and haze were analyzed using a haze meter (NDH-2000N, Nippon Denshoku, Japan). The transmittance, parallel transmittance, diffuse transmittance, and haze of translucent wood were measured and calculated according to ASTM D1003. Specimens used to measure transmittance were prepared in a size of 40 × 30 × 2 mm^3^.

### 2.8. Bending Strength Measurement

Bending strength was measured using UTM (Instron 4482, Instron, Norwood, MA, USA). The bending strength measurement sample was prepared in a size of 40 × 30 × 2 mm^3^. The bending strength test was conducted with a three-point load in two equal parts, and the strength of each cross-section and tangential cross-section was measured. Experiments were conducted until fracture occurred at a rate of 1 mm/min for untreated wood and 3 mm/min for epoxy-impregnated wood.

## 3. Results

### 3.1. Visual Inspection of Wood Specimens

The structure of wood is anisotropic; therefore, the properties vary according to the directionality of cell walls. Thus, the specimens were divided into a cross-section and tangential section in this study. To visually compare the wood states according to the treatment, images were taken of original wood (OW); cross-section (OWc), tangential section (OWt), and delignified wood (DW); cross-section (DWc), tangential section (DWt), and translucent wood (TW); cross-section (TWc) and tangential section (TWt), which are presented in Figure 1. The color of OW was yellowish gray with the rings clearly visible, while the color of DW was whitish gray with fading wood patterns. Placing the wood after transparent polymer impregnation on typed letters showed that the material was more translucent than transparent, as the letters appeared to be blurry. The produced wood in this study was thus termed as translucent wood (TW). Variations were found between the wood sections. For TW, the letters appeared to be blurrier under TWc than TWt, with visible wood patterns.

Meanwhile, the color of TW following epoxy impregnation was yellowish gray with an increase in yellowness over time. Similar results were reported by other studies. Cai et al. (2021) found that the yellowness of TW increased after the addition of ethylene glycol to the epoxy-impregnated wood to produce flexible TW [15]. The study showed that *Tilia americana* was delignified and poly methyl methacrylate (PMMA) was impregnated and then irradiated with UV-C (250 nm), and within a few hours of irradiation, discoloration of residual lignin towards yellow was observed with reduced transmittance [16]. The other study used *Pinus sylvestris* and *Prunus serotina* for delignification and epoxy impregnation to produce TW, and the TW kept in an outdoor condition was found to have undergone a color change towards yellow [17].

### 3.2. Functional Group Analysis

To examine the chemical changes on the wood surface, the changes in functional groups of wood and epoxy were determined (Figure 2a), and the contents of cellulose, hemicellulose, and lignin were measured to identify the chemical composition (Figure 2b). The Fourier-transform infrared spectroscopy (FT-IR) result showed that the lignin and hemicellulose peaks (1726, 1635, 1591, 1509, 1240 cm^−1^) were either reduced or removed at values in the range 2000–800 cm^−1^ after delignification. It was indicative of the role of peracetic acid in removing lignin and a portion of hemicellulose, which has been reported in many studies. Ma et al. (2016) showed that peracetic acid degraded the C-C bond and β-O-4 linkage of lignin for perfect depolymerization even at low concentrations [18]. In addition, Park et al. (2019) showed that peracetic acid, generated by mixing varying concentrations of acetic acid and hydrogen peroxide, could degrade lignin at room temperature [19]. In this study, an increase in acetic acid concentration used to produce peracetic acid led to a greater level of lignin degradation, but with a fall in the treatment time, hydrogen peroxide assisted with lignin degradation by acetic acid. The study showed that peracetic acid was used in delignifying a hardwood and softwood in varying time conditions, and not only lignin but also hemicellulose was shown to have been removed [20]. The treatment of a hardwood with peracetic acid for five hours showed that xylan and arabinan contents steadily decreased when the sugar yield was checked every hour. The variation in chemical composition according to the peracetic acid treatment showed that, after an eight-hour treatment with peracetic acid, *P. tomentosa* had approximately 0.93% residual lignin; on the contrary, *O. pyramidale* had approximately 2% residual lignin [21]. The result indicated that *P. tomentosa* and *O. pyramidale* shared similar rates of delignification, although variations may arise according to the condition of the solution used.

Meanwhile, the FT-IR result according to the graph for epoxy-impregnated TW showed a similar behavior as that of epoxy. This is due to the formation of a thin epoxy layer on the surface of TW as epoxy filled the pores which had increased in size by delignification [22].

### 3.3. Changes in Cell Wall Structures

Scanning electron microscope (SEM) images were taken of the untreated, delignified, and epoxy-impregnated wood specimens of *P. tomentosa* between cross- and tangential sections, which are presented in Figure 3. The images of untreated wood showed that all cellular lumens were empty. In the tangential section, a medullary ray was observed, while in the cross-section, tylosis was found as in the wood fibers of hardwoods. The tangential section of delignified wood showed certain areas of destroyed cell walls compared to untreated wood, while the overall wood structure was intact with empty lumens. The cross-section also showed empty spaces in intercellular layers that led to an increased size of pores. The SEM result of translucent wood showed that polymer impregnation at 0.1 MPa had not destroyed the micro-structure of wood; in fact, the structure was shown to have been strengthened via resin impregnation. However, a few small cracks were observed in the area where the cell wall came into contact with the epoxy found inside the cell wall to imply possible incomplete filling of pores by epoxy.

According to Wu et al. (2019), cracks were observed between the cell wall and the resin on the cross-section of wood after PMMA impregnation, which was reported to indicate the imbalance of binding according to morphological differences between the two polymers [23]. According to Li et al. (2017) and Wang et al. (2021), small cracks were observed between the cell wall and the resin, which was reported to be due to the formation of a layer of air to scatter the light to reduce translucency and increase the haze [8,17]. On the tangential section, impregnation was shown to have led to the filling in general, without any medullary ray being detected.

### 3.4. Transmittance by Section

The transmittance and haze of translucent wood and epoxy were measured, and the results are presented in Table 1. Epoxy was chosen as the polymer for impregnating translucent wood not only because of its excellent mechanical strength but also because of its refractive index, which is similar to that of common glass (~1.5 of glass vs. 1.52 of epoxy) [24]. To compare the transmittance and haze according to the directionality of wood, the epoxy used in impregnation was set as the control. The total transmittance of epoxy was 91.46 ± 0.23%, and the parallel transmittance was high at 87.11 ± 0.38%, while the diffuse transmittance and haze were low at 4.35 ± 0.24% and 4.76 ± 0.26%, respectively. As such, translucency may be ensured when the material exhibits high levels of total transmittance and parallel transmittance but low levels of diffuse transmittance and haze. On the other hand, the cross-section of translucent wood had low parallel transmittance at 3.46 ± 0.19% and high diffuse transmittance at 87.80 ± 0.74%, with high total transmittance at 91.25 ± 0.81%, while the haze was also considerably high at 96.21 ± 0.19%. The tangential section had parallel transmittance at 19.06 ± 11.29% and diffuse transmittance at 68.92 ± 9.69%, with total transmittance at 87.98 ± 1.60% and the haze at 78.48 ± 12.32%. The parallel transmittance of the tangential section was higher than the cross-section by 15.60% to allow for marginally higher visibility, but the haze and diffuse transmittance were high.

For translucency, there should be no other component exhibiting a deviating refractive index such as a layer of air on the interior of the material, while amorphous areas should be abundant [25]. As can be seen in Figure 3l, however, translucent wood had a layer of air due to the small cracks between the cell wall and the resin, and the crystalline region of the cell wall led to high levels of dispersion and scattering [7]. The inner layer of air and crystalline region is associated with the thickness of the translucent wood. According to Chen et al. (2019), transmittance decreased as the wood thickness increased by 1 mm, and as acetylation proceeded, transmittance increased, and the haze decreased [11]. Qin et al. (2018) also reported a reduction in transmittance with an increase in wood thickness [26].

### 3.5. Bending Strength by Section

The measurements of bending strength of translucent wood by section are presented in Table 2, in which the strength was higher even after impregnation for the tangential section than for the cross-section. Based on the mechanical strength of epoxy used in impregnation, the bending strength increased for the translucent wood compared to that of the untreated wood of *P. tomentosa*. The modulus of elasticity (MOE) increased to 549.65 MPa for the cross-section and 1476.42 MPa for the tangential section, whereas the modulus of rupture (MOR) increased to 32.52 MPa for the cross-section and 39.79 MPa for the tangential section. The anisotropy of the wood structure and the resulting mechanical strength were shown to not have been altered through epoxy impregnation. It is thus conjectured that the properties of translucent wood follow those of the original wood. Epoxy impregnation increases the bending strength as the epoxy fills the cellular lumens and binds with the cell walls in the vicinity.

Similar results were reported in other studies. Li et al. (2018) stated that the cell wall and the polymer used in impregnation were the components that bear the load of translucent wood [13]. In Li et al. (2016), PMMA was used as the polymer in impregnation, and the tensile strengths of the translucent wood, PMMA, and delignified wood were measured [7]. The elastic modulus and stress were 3.6 GPa and 90.0 MPa for translucent wood with PMMA impregnation, 1.8 GPa and 44.0 MPa for PMMA, and 0.2 GPa and 3.0 MPa for delignified wood, respectively, which indicated a substantial increase in the tensile strength. In addition, Lang et al. (2018) measured the bending strengths of translucent wood and glass and showed that the strain up to destruction was 0.20 ± 0.02% for glass and 2.9 ± 0.2% for translucent wood [27]. The MOR was 98 ± 7 MPa for untreated *Betula platyphylla*, 140 ± 10 MPa for translucent wood, 72 ± 6 MPa for PMMA, and 116 ± 13 MPa for glass, so that the bending strength was reported to have increased for the translucent wood. Meanwhile, epoxy impregnation was shown to have altered the property of wood; wood is originally a brittle material, but the wood after epoxy impregnation exhibited some elasticity. The results were similar across PMMA and other polymers [13,17,28,29].

## 4. Conclusions

This study investigated the production of translucent wood from *P. tomentosa*. The main findings are summarized as follows:

The diffuse transmittance and haze were 87.80 ± 0.74% and 96.21 ± 0.19% for the cross-section and 68.92 ± 9.69% and 78.48 ± 12.30% for the tangential section, respectively. The directionality of cell walls influenced the directionality of the light dispersion along the cell walls and the variations in the distribution of cracks to ultimately induce variations in optical properties between the sections.

Untreated wood showed an MOR of 8.70 ± 0.77 MPa on the cross-section, which increased to 41.22 ± 3.79 MPa after impregnation, and of 56.90 ± 16.45 MPa on the tangential section, which increased to 96.99 ± 14.22 MPa after impregnation. The anisotropy of the cross-section and tangential section of the original wood and the resulting bending strength were maintained in the translucent wood, which highlights the importance of selecting the section with a high level of bending strength in applying the wood material as a structural component. As noted in the results and discussion section, translucent wood made of *P. tomentosa* showed similar performances on bending strength, transmittance, and haze compared to that of balsa. This suggested that Paulownia wood has great potential as a substitute for that wood.

After epoxy impregnation, small cracks formed due to the difference in physical properties between the cell walls and epoxy, which suggested the formation of an inner layer of air. In the transmission of light, the different refractive indices of the layer of air, the crystalline regions in cell walls, and the epoxy led to high levels of diffuse transmittance and haze of the translucent wood. In line with this, the visibility was low unless the target was in contact with the wood to prevent clear observations. Therefore, to decrease the haze and increase the transmittance, further studies should explore the methods of wood pre-processing so as to turn the crystalline regions of cell walls into amorphous regions and to reduce cracks that occur upon polymer hardening. Studies on insulation should also be conducted with a focus on the scattering properties of direct light.

Translucent wood with epoxy impregnation showed yellow discoloration with time. If lignin is not completely removed by delignification, the residual lignin and bisphenol-A used in the epoxy synthesis are presumed to cause discoloration over time.

The final goal of this study is to change glass or transparent plastics to all-biodegradable transparent or translucent materials. As basic research, translucent wood was fabricated using commercial epoxy resin. Next, research is being planned to conduct a study to increase the transparency of the translucent wood and to make all-biodegradable materials using biopolymers. 

## Figures and Tables

**Figure 1 polymers-14-04380-f001:**
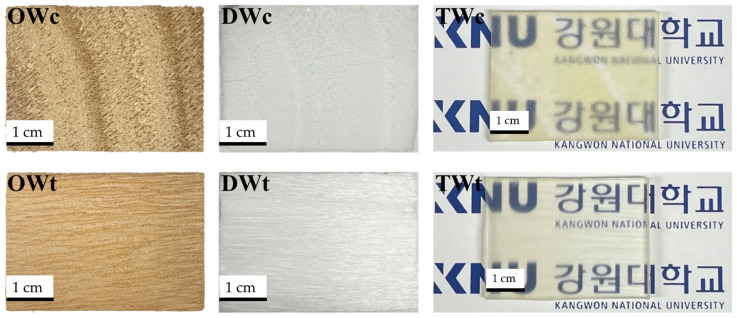
Images of OWc (original wood cross-section), DWc (delignified-wood cross-section), TWc (translucent wood cross-section), OWt (original wood tangential section), DWt (delignified-wood tangential section), and TWt (translucent wood tangential section).

**Figure 2 polymers-14-04380-f002:**
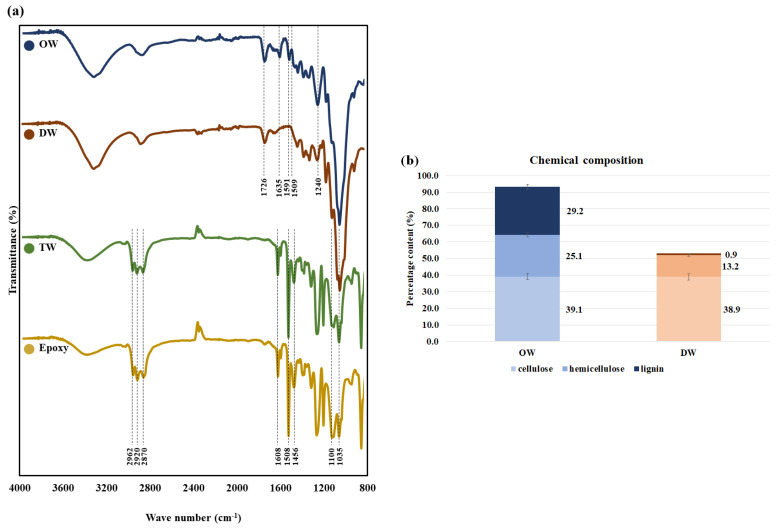
Changes in chemical properties by delignification. (**a**) Cellulose, hemicellulose, and lignin content of OW and DW. (**b**) FT-IR spectra of OW, DW, TW, and epoxy.

**Figure 3 polymers-14-04380-f003:**
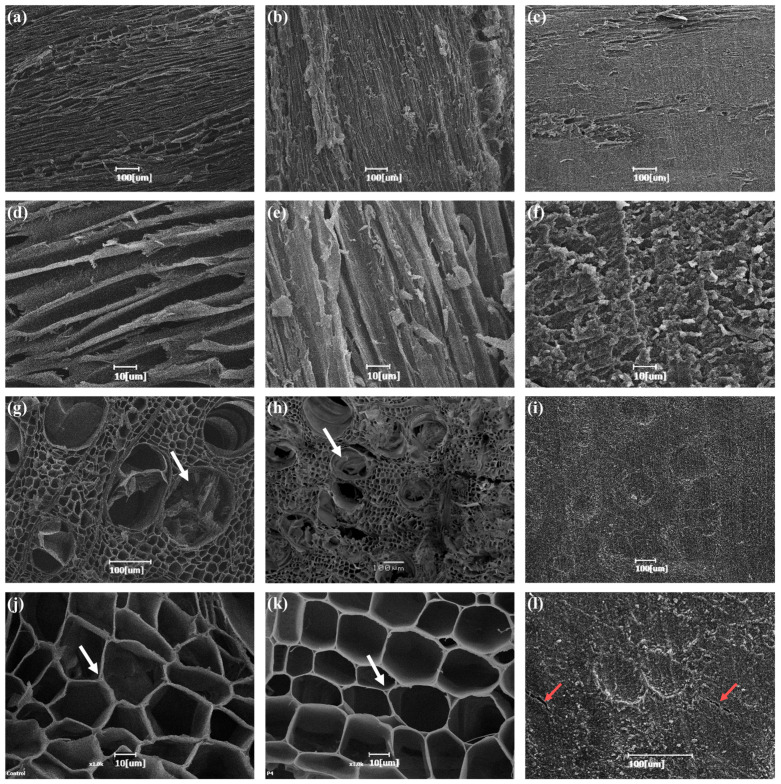
SEM images of *Paulownia* wood and transparent woods. Original wood (**a**,**d**,**g**,**j**), delignified wood (**b**,**e**,**h**,**k**), and translucent wood (**c**,**f**,**i**,**l**).

**Table 1 polymers-14-04380-t001:** Transmittance and haze of epoxy and translucent wood.

	Total Transmittance (%)	Parallel Transmittance (%)	Diffuse Transmittance (%)	Haze (%)
Epoxy	91.46 (±0.23)	87.11 (±0.38)	4.35 (±0.24)	4.76 (±0.26)
TWc	91.25 (±0.81)	3.46 (±0.19)	87.80 (±0.74)	96.21 (±0.19)
TWt	87.98 (±1.60)	19.06 (±11.29)	68.92 (±9.69)	78.48 (±12.32)

**Table 2 polymers-14-04380-t002:** The average bending strength values of untreated and translucent wood.

	OWc	TWc	OWt	TWt
MOE (MPa)	856.80 (±277.58)	1406.45 (±284.54)	4613.47 (±1636.48)	6089.88 (±1207.09)
MOR (MPa)	8.70 (±0.77)	41.22 (±3.79)	56.90 (±16.45)	96.99 (±14.22)

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
