# Peer review of "Characterization of a Translucent Material Produced from Paulownia tomentosa Using Peracetic Acid Delignification and Resin Infiltration"

_polymers, 2022, doi:10.3390/polym14204380_

Round 1
Reviewer 1 Report
In my opinion, the article: "Characterization of a translucent material produced from Paulownia tomentosa using peracetic acid delignification", deserves to be published after a minor revision. English language has to be slightly improved in few sections, mainly using the correct technical terms to describe the investigated properties of the materials dealt with. From a scientific base point the experimental part has been quite appropriately carried out, despite improvements are mandatory. Results are quite clearly presented and the related interpretation can be agreed, even if some improvements are mandatory also from this point of view.
Here some question:
Abstracts. Raws 15-18. The authors claim: " For the bending strength, untreated wood showed values of approximately 4613.5 MPa modulus of elasticity (MOE), while the epoxy impregnation to improve the strength of wood had increased MOE up to approximately 6089.9 Mpa, respectively."
- Epoxy matrix, as claimed by the authors, is commonly said in the field of composite materials. The related binding and stabilization performance is higher than hemicellulose and lignin, also in the case of cellulose fibers. For this reason it appears quite normal that mechanical properties of epoxy wetted materials is higher than the natural wood ones. In my opinion, this test is not sufficient (or doesn't represent the best solution) to set off the effectiveness of the wood treatment carried out;
- Paragraph 2.6. The authors revealed size and measures of the specimens used to carry out three point bending test. Did the authors follow any standard to carry out this experimental part? (i.e. ISO, ASTM, or anything else);
- The authors used two testing speed for the two different kinds of material. In my opinion this could negatively affect the scientific value of this experimental part, in particular in terms of result comparability. For this reason they are invited to specify the reason for this choice and to demonstrate that it is not a critical issue for the result comparability;
- Pag 7/10. Raws 227 - 231. Measured value of the transmittance have be dealt with. In some case high standard deviation have been found. The authors invited to better explain/justify this aspect;
- Pag 7/10. The authors use the term: "MOR - Modulus of Rupture". Did they mean Breaking Strenght?. In my opinion the use of the latter term could be better from a scientific point of view;
- Pag 8/10. The authors appropriately summarized the measured mechanical properties in a table. In my opinion this is not sufficient to allow the reader understanding the actual differences among the investigated materials. Stress-strain curves of the mentioned materials could be presented also in the same diagram to support the discussion of the results. This, also because the authors mention the materials "malleability" (raw 272). This term is not so commonly used (or appropriate) in the field of plastic and composite materials. On this regards, in the text it is not clear if it is referred to deformation capability in the elastic or in the plastic field of the materials, despite composite materials generally don't show significant plastic deformation.
The authors are invited to change the mentioned term with "elastic or plastic deformation capability" (or something similar). Furthermore, the values of the breaking strain have to be summarized in table 2, together with the other properties;

Author Response
Thank you for your very careful review of our paper. Please see the attachment.

Reviewer 2 Report
The authors present an interesting article entitled "Characterization of a translucent material produced from Paulownia tomentosa using peracetic acid delignification".
The abstract is ok.
The introduction is ok.
The experimental section is ok.
The results and discussion section is ok.
The conclusion is ok.
The references are ok.
Please change the title from "Characterization of a translucent material produced from Paulownia tomentosa using peracetic acid delignification" to read "Characterization of a translucent material produced from Paulownia tomentosa using peracetic acid delignification and resin infiltration"
Introduction:
References "Li et al. (2016)" should be numbered.
2.1.
The authors express dimensions as: LxRxT - perhaps more accurately LxWxT (length x width x thickness)?
When reporting dimensions, mm3 should be mm2.
2.2.2.
Please clarify the ratio of wood:delignification agent (perhaps also include a schematic in the supplementary information).
Please clarify any sample names generated by this process.
2.3.
Please clarify the ratio of wood:epoxy resin (perhaps also include a schematic in the supplementary information).
Please clarify any sample names generated by this process.
2.4. please clarify how the samples were coated by Pt (I guess sputter coating) and the thickness of the Pt layer?
2.5. when reporting dimensions, mm3 should be mm2.
2.6. when reporting dimensions, mm3 should be mm2.
3.1.
In the 1st paragraph, please clarify any sample names generated by the processes described in the experimental section by inclusion of a table detailing sample names & sample preparation conditions.
References (e.g. Cai et al. (2021)) should be numbered.
Figure 1: needs scale bars - the legend also needs to clarify this.
3.2.
References (e.g. Ma et al. (2021)) should be numbered.
3.4.
Sample names in Table 1 need to be clarified - should there be errors (e.g. 91.46 +/- X)? should the numbers really be reported with 2 decimal places?
3.5.
Sample names in Table 2 need to be clarified - should there be errors (e.g. 91.46 +/- X)? should the numbers really be reported with 2 decimal places?
XRD data would be interesting and should support the mechanical data.
Author Response

(The authors gave the same response as above.)

Reviewer 3 Report
Comments:
This study deals with the characterization of a translucent materials produced from paulownia wood using a peracetic acid delignification approach. The results are not completely new and did not bring a new insight in this topic. The presentation of the results does not correspond to this journal with a high impact factor. Therefore, manuscript cannot be recommended for publishing the in Polymers at the current stage.
Introduction
The red thread is missing. Please also add some important details about the previous studies;
Line 55 to 67: “Meanwhile, the Intergovernmental Panel on Climate Changes (IPCC) guideline … with respective countries [12].” The delignification process and the impregnation with epoxy resin do not results in CO2 storage. Why is the mentioned behaviour so important for that manuscript? Did you have some results from LCAs?
Line 57 to 66: “Amount the well-known … high porosity.” What are the main challenges for the delignification processes? It is not always a problem for wood from South Korea or Asia? Please add the units! What did you mean with “high porosity”?
Line 66 to 70: “In this study, delignification …. Material as an alternative to glass.” What was the scientific objective of this study? You use only one wood species and one method for the delignification process as well as one impregnation method. Therefore, you cannot evaluate the delignification or even the impregnation process in detail.
Materials and Methods
Line 94: “… stored in acetone for acetylation”. Please could you describe the acetylation process and the used materials?
Please add details of the FT-IR Spectroscopy and the method for analysing the chemical compositions of the wood (e.g. Figure 2a);
3. Results
Line 158 to 159: “… that the lignin and hemicellulose peaks …” What did you mean with the lignin and hemicellulose peaks? Please add some details;
Line 162: “… the C-C bond and … of lignin …” What is the difference between C-C bond of lignin or Cellulose? What did you mean C=C or other one group/vibration (e.g. aromatic skeletal vibration)?
How did you analyse the chemical composition (Figure 2a)?
Figure 3: The quality of the Figure 3 has to be improved.
Table 1: What did you mean with the terms “Translucent Wood-C” and “Translucent Wood-T” (wood-c and wood-t) in Table 1?
Conclusion
The conclusion section did not represent all parts of this manuscript. Only the mechanical part was concluded and further studies and aims were mentioned.
Author Response

(The authors gave the same response as above.)

Round 2
Reviewer 3 Report
The authors told us that the final goal of this study (conclusion section) was to alternate glass or transparent plastics to all biodegradable transparent or translucent materials as well as the storage of CO2 emissions. However, I wonder about the process of achieving their goals. If you wrote these things, then the readers of this manuscript could have different feelings of the research directions. The authors used standard chemicals and processes told us about the impact mitigation for environment. However, the authors did not show any results of CO2 emission, biodegradable polymers, or even innovative processes instead of standard processing processes. Which process has the minimal impact to the environment (i) wood with delignification process and standard resin or ii) standard polymers)? Without adding such new approaches, the paper has not a big impact for the scientific community. In general, the red thread is missing.
For example:
Line 19-20: “A comparative analysis was performed in this study with respect to the substitution of balsa, which is used widely in the production of translucent wood.” However, in the conclusion section, the reader cannot find any comparative conclusion about the substitution of balsa.
Line 56-58: “... the storage effect of delayed carbon emission via the consumption of harvested wood products …" The reader of this manuscript will have more information and other feelings, but you wrote one sentence concerning this topic and no more. Why is it important for your study?
Line 64-66: “In this study, delignification was applied to P. tomentosa, and the potential of translucent wood was evaluated by impregnating the pores with increased size accordingly with transparent epoxy.” This was the real aim of your study. However, you used only one wood species, one delignification process and one standard epoxy resin. Why did you tell the readers that things about balsa wood, biodegradable, and CO2 emissions?
Author Response
Please see the attachment. Thank you for your review.

Round 3
Reviewer 3 Report
The authors reworked the manuscript. The editor should make the decision.